# Understanding Compositional Generalization via Hierarchical Concept Models

## Abstract

Compositional generalization – the ability to understand and generate novel combinations of learned concepts – enables models to extend their capabilities beyond limited experiences. While humans perform this task naturally, we still lack a clear understanding of what theoretical properties enable this crucial capability and how to incorporate them into machine learning models. We propose that compositional generalization fundamentally requires decomposing high-level concepts into basic, low-level concepts that can be recombined across similar contexts, similar to how humans draw analogies between concepts. For example, someone who has never seen a peacock eating rice can envision this scene by relating it to their previous observations of a chicken eating rice. In this work, we formalize these intuitive processes using principles of causal modularity and minimal changes. We introduce a hierarchical data-generating process that naturally encodes different levels of concepts and their interaction mechanisms. Theoretically, we demonstrate that this approach enables compositional generalization supporting complex relations between composed concepts, advancing beyond prior work that assumes simpler interactions like additive effects. Furthermore, we show that the true latent hierarchical model can be recovered from data under weaker conditions than previously required. By applying insights from our theoretical framework, we achieve significant improvements on benchmark datasets, verifying our theory.

## 1 Introduction

Compositional generalization is a hallmark of human intelligence, enabling us to navigate a vast array of novel situations despite limited direct experience. This capability is particularly evident in visual recognition – someone who has never seen a peacock eating rice can readily visualize such a scene based on separate exposures to peacocks and rice. This ability clearly depends on favorable structures in the underlying data distribution. If peacocks consume rice in a manner entirely unlike other observed feeding behaviors, one cannot expect to accurately visualize this unfamiliar scene. In light of this, we aim to address the following fundamental question:

*What data structures enable compositional generalization, and how can we characterize them?*

Answering this question is essential for deliberately incorporating this valuable capability into machine learning models, which typically perform poorly when confronted with data outside their training distribution (Koh et al., 2021; Recht et al., 2019; Taori et al., 2020). Despite substantial empirical advances (Ramesh et al., 2022; Du & Kaelbling, 2024; Liu et al., 2022; Zhang et al., 2024c; Hu et al., 2024; Huang et al., 2023; Bar-Tal et al., 2023; Yang et al., 2024b), theoretical understanding remains limited and often relies on restrictive assumptions about concept interactions. For instance, Brady et al. (2023) and Wiedemer et al. (2024a) assume concepts affect separate pixel regions without interaction, while Lachapelle et al. (2023) proposes additive concept influences in pixel space, later extended by Brady et al. (2024) to include second-order polynomial terms. Importantly, these contributions generally overlook the hierarchical nature of concepts and their relationships in latent space, limiting their ability to capture the richness of real-world image distributions.

In this work, we draw inspiration from humans' cognitive process of *drawing analogies*, which achieves compositional generalization by comparing and relating observed concepts. In particular, this mechanism entails two steps: 1) dissembling complex, high-level concepts into low-level

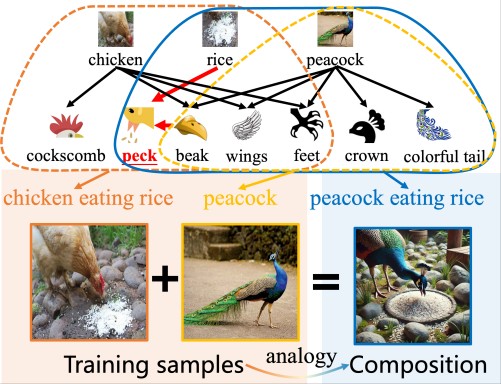

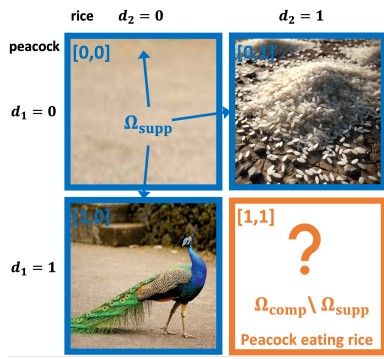

Figure 1: **Compositional generalization.** The hierarchical process generalizes to an unseen concept "peacock eating rice" by composing low-level modules "peacock" and "chicken eating rice". The interaction "beak & rice" (denoted as "peck") transfers from "chicken eating rice" to "peacock eating rice", akin to human making analogies.

Figure 2: **Compositional generalization.** The training support $\Omega_{\text{supp}}$ only contains "peacock" and "rice" separately. Compositional generalization aims to generate "peacock eating rice" in the out-of-support region $\Omega_{\text{comp}} \setminus \Omega_{\text{supp}}$.

ones, and 2) recombining these low-level concepts to represent interactions in novel scenarios. As illustrated in Figure 1, even without having seen a peacock eating rice, one can relate this scene to observations of a chicken eating rice by recognizing shared low-level features—both peacocks and chickens have beaks, wings, and other common attributes. This process effectively transforms interactions between high-level concepts (peacock-rice) into interactions between more fundamental low-level concepts (beak-rice). Since these elementary interactions appear across many observed scenarios, they can be transferred to accurately visualize novel combinations of high-level concepts.

We formulate these intuitions in the language of causality, specifically, a latent *hierarchical* model that encodes *causal modularity* and the *minimal-change* principle. Causal modularity enables the transfer of modular low-level interactions (e.g., "beak & rice"). Meanwhile, the minimal-change principle explains that some high-level concepts share many low-level features while being distinct in a few, making knowledge transfer possible. Unlike prior work (Wiedemer et al., 2024a; Lachapelle et al., 2023; Brady et al., 2024; 2023), our framework accommodates complex interactions among high-level concepts and their intricate relations in the latent space.

We establish identification conditions that allow latent hierarchical models to be recovered from observed data, e.g., text-image pairs ubiquitous in image generation tasks. Unlike previous work on identifying latent hierarchical models (Huang et al., 2022; Choi et al., 2011; Anandkumar et al., 2013; Pearl, 1994; Dong et al., 2024), our theory does not require linearity or discrete latent variables, making it capable of modeling more complex data distributions. While recent work (Kong et al., 2023) also addresses nonlinear hierarchical models and identifies concepts in groups, our approach leverages interactions among latent variables to identify *individual* latent concepts and the graphical structure. Building on this theoretical foundation, we demonstrate how the abstract concepts of hierarchical levels and modularity can be practically realized by interpreting diffusion timesteps as hierarchical levels and by enforcing an explicit sparsity regularizer on concept attention maps. Our empirical results validate that integrating these theoretically-motivated design choices leads to significant improvements in compositional generation.

Please refer to Appendix A for a more detailed discussion on related work.

## 2 COMPOSITIONAL GENERALIZATION AND HIERARCHICAL MODELS

In this section, we formally define compositional generalization and introduce the latent hierarchical data-generating process, which lies at the core of our framework. We denote the dimensionality of a multidimensional variable with $n(\cdot)$, the integer set $\{i\}_{i=1}^{n}$ with $[n]$, and all parents of $v$ with $\text{Pa}(v)$.

**Compositional generalization.** Let $\mathbf{x}$ denote the observed variables $\mathbf{x} \in \mathbb{R}^{d_{\mathbf{x}}}$ of interest (e.g., natural images). Let $\mathbf{d} \in \{0, 1, \dots\}^{n(\mathbf{d})}$ be discrete variables that control high-level concepts present in the paired data $\mathbf{x}$ (e.g., "peacock"). Then, text-to-image generation entails learning the condition

distribution $p(\mathbf{x}|\mathbf{d})$, where we specify the discrete concepts $\mathbf{d}$ through text to generate the corresponding image $\mathbf{x}$. However, the training data distribution often lacks data containing certain combinations of concepts, even when each concept appears separately. In Figure 2, $d_1$ and $d_2$ indicate the presence of "peacock" and "rice" when they take value 1. Although we may observe "peacock" and "rice" in separate images (i.e., the training distribution contains images with $\mathbf{d} = [0, 1]$ and $\mathbf{d} = [1, 0]$), their composition $\mathbf{d} = [1, 1]$ required to produce "peacock eating rice" may be absent from the training support $\Omega_{\text{supp}}$. Since we can only train the model $\hat{p}(\mathbf{x}|\mathbf{d})$ to match the true distribution $p(\mathbf{x}|\mathbf{d})$ over the support $\{[0, 0], [0, 1], [1, 0]\}$, the model $\hat{p}(\mathbf{x}|\mathbf{d})$ might produce arbitrary results for the out-of-support input $\mathbf{d} = [1, 1]$. In this context, *compositional generalization* refers to when our model $\hat{p}(\mathbf{x}|\mathbf{d})$, which agrees with the true model $p(\mathbf{x}|\mathbf{d})$ on the support $\Omega_{\text{supp}}$, agrees on a strictly larger space $\Omega_{\text{supp}} \subset \Omega_{\text{comp}}$. We call a set of concepts $\mathbf{d}$ *composable* if it lies within the compositional space $\mathbf{d} \in \Omega_{\text{comp}}$. An important example of $\Omega_{\text{comp}}$ is the Cartesian product space $\Omega_{\text{CP}} := [\Omega_{\text{supp}}]_1 \times \cdots \times [\Omega_{\text{supp}}]_{n(\mathbf{d})}$ (Lachapelle et al., 2023; Wiedemer et al., 2024b) where $[\Omega_{\text{supp}}]_i := \{d_i : d \in \Omega_{\text{supp}}\}$ denotes the marginal support of dimension $i$. In this case, the model should correctly compose concepts that appear separately in the training.

**Challenges and motivations.** Recent work in causal representation learning has increasingly focused on establishing provable conditions for compositional generalization. To address the extrapolation challenge, prior work (Lachapelle et al., 2023; Wiedemer et al., 2024b;a) proposes additive generating functions, where the joint influence of multiple latent concepts $\mathbf{z}_i$ can be expressed as the sum of their individual influences $\mathbf{x} := \sum_i g_i(\mathbf{z}_i)$. While this semi-parametric approach offers certain compositional properties, it fails to adequately model complex interactions among concepts, as it limits all concept interactions to mere addition of their individual pixel values. More recently, Brady et al. (2024) leverage interaction asymmetry properties to partially overcome this limitation. However, their approach still characterizes concept interactions using a restrictive parametric form (polynomials), which may not capture the full range of complex interactions in real-world data. Thus, it remains a significant challenge to identify natural properties in the data-generating process that can support general concept interactions.

**Causal modularity and minimal changes.** Humans understand and envision concept compositions through *comparison* and *analogy*, cognitive processes that align with causal principles of *causal modularity* and *minimal changes*.

*Causal modularity* refers to how high-level concepts decompose into transferable low-level modules. The concept "peacock" breaks down into components (i.e., low-level concepts) like "beak", "wings", and "colorful tail," while "chicken" decomposes into "beak", "wings," and "cockscomb" (Figure 1). These components function as modular building blocks that can be recombined across contexts. The interaction patterns between these components are also transferable—the mechanism by which a "beak" interacts with "rice" forms a reusable module applicable across different bird species. This architecture enables efficient representation of complex concepts for humans.

The *minimal-change* principle complements modularity by emphasizing that high-level concepts largely share low-level concepts, with only minimal distinguishing features. When comparing "peacock" and "chicken", both activate many identical low-level concepts (e.g., "beaks", "wings"), differing primarily in specific attributes ("colorful tail" vs. "cockscomb"). This overlap facilitates comparison and analogy between related concepts. We intuitively recognize peacocks and chickens as more similar to each other than to fish precisely because they share more low-level concepts.

Together, these properties empower humans to envision novel concept combinations never directly experienced. Consider "peacock eating rice" as a novel combination (Figure 1). We can readily imagine this because: (1) modularity allows decomposition of "peacock" into components including a "beak" and enables transfer of the interaction module "beak & rice" observed in "chicken eating rice", and (2) minimal changes enables recognition that despite differences in appearance (e.g., "colorful tails"), peacock beaks share properties with chicken beaks that interact with rice similarly.

This framework explains our ability for compositional generalization – we decompose high-level concepts into transferable low-level modules and leverage the shared features for analogical reasoning, while accounting for minimal distinguishing features that preserve conceptual uniqueness.

**Hierarchical data-generating processes.** To encode these key properties, we formulate a hierarchical data-generating process to explicitly model concepts at distinct hierarchical levels and their

interactions. Let latent variables be $\mathbf{z} := [\mathbf{z}_1, \cdots, \mathbf{z}_L]$, where $L$ denotes the total number of hierarchical levels and $\mathbf{z}_l \in \mathbb{R}^{n(\mathbf{z}_l)}$ represents concept variables on the hierarchical level $l \in [L]$. We denote the hierarchical graphical model as $\mathcal{G} :=$ $(\mathcal{V}, \mathcal{E})$ where $\mathcal{V} := \mathbf{d} \cup \mathbf{z} \cup \mathbf{x}$ denotes the variable set and $\mathcal{E}$ denotes the edge set. [1] We present the data-generating process in equation 1 and Figure 3.

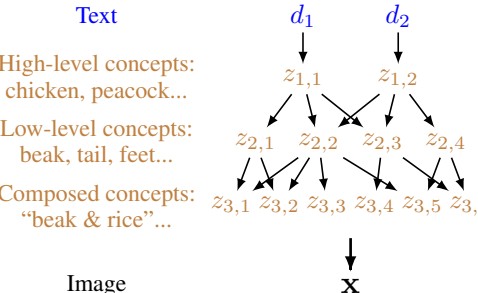

$$z_{1,i} \sim p(z_{1,i}|d_i), \quad v := g_v(\mathrm{Pa}(v), \epsilon_v), \quad (1)$$

where $v \in \mathcal{V} \setminus (\mathbf{z}_1 \cup \mathbf{d})$ represents all non-root variables and $\epsilon_v$ denotes its independent exogenous information. The discrete variable $\mathbf{d}$ (i.e., textual descriptions) directly specifies high-level concepts $\mathbf{z}_1$ (e.g., their presence or specific categories). For instance, $d_1 = 0$ may indicate the absence of "peacock", while $d_{1,1} = 1$ and $d_{1,1} = 2$ might signify two varieties of "peacocks". We assume that the conditional distribution $p(z_{1,i}|d_i = 0)$ is degenerate (i.e., a constant $z_{1,i}$) to indicate its absence and has identical $\mathrm{supp}(p(z_{1,i}|d_i))$ when $d_i$ takes on different

Figure 3: **A hierarchical data-generating process.** We denote the textual description as $\mathbf{d}$, continuous latent concepts as $\mathbf{z}$, and the image as $\mathbf{x}$, where text-image pairs $p(\mathbf{d}, \mathbf{x})$ are observable.

nonzero values to represent different varieties of the same concept $z_{1,i}$. We denote model parameters as $\boldsymbol{\theta} := \big( p(\mathbf{z}_1, \mathbf{d}), \{g_v\}_{v \in \mathcal{V} \setminus (\mathbf{z}_1, \mathbf{d})} \big)$. For exposition, we refer to $\mathbf{x}, \mathbf{d}$ as $\mathbf{z}_{L+1}, \mathbf{z}_0$ respectively.

## 3 COMPOSITION CONDITIONS AND IDENTIFIABILITY

We first demonstrate how compositional generalization can arise from the hierarchical data-generating process present in natural data (Section 3.1). Then, we show that one can learn such data-generating processes from image-text data $p(\mathbf{x}, \mathbf{d})$ under proper assumptions (Section 3.2).

### 3.1 COMPOSITIONAL GENERALIZATION CONDITIONS

**Remarks on the problem.** Although the variables $\{\mathbf{z}_l\}_{l \in [0, L+1]}$ form a Markov chain over $\mathbf{z}_l$, the first module $p(\mathbf{z}_1|\mathbf{d})$ could give rise to distinct supports $\mathrm{supp}(\mathbf{z}_1|\mathbf{d})$ across various values of $\mathbf{d}$ (i.e., missing concepts). In Figure 2, the training support $\Omega_{\mathrm{supp}}$ lacks the combination "peacock eating rice" $\mathbf{d} = [1, 1]$, which affects $\mathrm{supp}(\mathbf{z}_1)$ and propagates downstream through $\mathrm{supp}(\mathbf{z}_l)$ for $l \in [L+1]$. As children of $\mathbf{z}_1$, variables $\mathbf{z}_2$ only take on values from a more restricted set. Consequently, the matching between two models $\boldsymbol{\theta}$ and $\hat{\boldsymbol{\theta}}$ is only partially supported due to the incompleteness of $\Omega_{\mathrm{supp}}$. In Theorem 3.1, we characterize the extent to which the matching on the incomplete support $\Omega_{\mathrm{supp}}$ can generalize, thanks to the hierarchical model equation 1.

**Theorem 3.1** (Composition Generalization). *We assume the data-generating process equation 1. The discrete concept combination $\mathbf{d}$ is composable (i.e., $\mathbf{d} \in \Omega_{\mathrm{comp}}$) if for each continuous latent variable $z \in \mathbf{z}$, its parents' distribution support $\mathrm{supp}(\mathrm{Pa}(z)|\mathbf{d})$ is contained by $\mathrm{supp}(\mathrm{Pa}(z)|\tilde{\mathbf{d}})$ for some combination $\tilde{\mathbf{d}} \in \Omega_{\mathrm{supp}}$ on the support, i.e., $\mathrm{supp}(\mathrm{Pa}(z)|\mathbf{d}) \subseteq \mathrm{supp}(\mathrm{Pa}(z)|\tilde{\mathbf{d}})$.*

**Hierarchical structures benefit compositional generalization.** The key insight is that $\tilde{\mathbf{d}} \in \Omega_{\mathrm{supp}}$ can be chosen specifically for each latent variable $z \in \mathbf{z}$.

Intuitively, if generating a composition $\mathbf{d} \in \Omega_{\mathrm{comp}}$ (e.g., "peacock eating rice") entails two variables $z_1$ (e.g., "beak & rice") and $z_2$ (e.g., "colorful tail"), we only require the two concepts $z_1$ and $z_2$ to appear *separately* in some supported discrete concepts $\tilde{\mathbf{d}}_1, \tilde{\mathbf{d}}_2 \in \Omega_{\mathrm{supp}}$. For instance, $\tilde{\mathbf{d}}_1$ and $\tilde{\mathbf{d}}_2$ could be "chicken eating rice" and "peacock". This is a direct application of *causal modularity* where we independently transfer and utilize the modules of $z_1$ and $z_2$ to create the novel combination $\mathbf{d} \in \Omega_{\mathrm{comp}}$. Thanks to the *hierarchical structure*, each observed variable $x$ (i.e., pixels) receives influences from multiple top-level concepts in $\mathbf{z}_1$ through a composition of nonparametric transformations along the hierarchical process, capable of capturing complex interactions of these concepts. This is because the hierarchical model allows for transferring low-level modules shared across distinct high-level concept combinations. For instance, we can utilize the low-level module of "beak

---

[1]We view multidimensional variables as *sets* when appropriate (e.g., $\mathbf{x}$ as $\{x_i\}_{i \in [d(\mathbf{x})]}$).

& rice" learned from the high-level concept "chicken eating rice" to render "peacock eating rice". Since these transferable low-level modules are learned from data, they are flexible enough to encode the natural interaction as needed. In contrast, prior work Wiedemer et al. (2024a); Lachapelle et al. (2023); Brady et al. (2024) only permits each $x$ to embody minimal concept interactions (i.e., parametric functions like additions), resulting in a trade-off between compositionality and representative power. In addition, the *minimal change* principle indicates that high-level concepts could share many low-level modules and differ only in a small set of concepts. In Figure 1, "chicken" and "peacock" share "beak" and "wings" and differ in a few concepts like "colorful tail". Consequently, only a small fraction of modules need to transfer, making the composition more plausible.

**Composability and sparsity.** Theorem 3.1 highlights the crucial role of the graphical model's *sparsity* for compositional generalization: a sparse model features smaller parental sets $\mathrm{Pa}(z)$ which impose fewer constraints for the module transfer. This makes it more likely to find a $\tilde{\mathbf{d}} \in \Omega_{\text{supp}}$ on the support that includes its parents' support $\mathrm{supp}(\mathrm{Pa}(z)|\mathbf{d})$ for each variable $z$. Thus, hierarchical models with sparser graphs offer stronger compositional capability. This insight naturally suggests sparse regularization during model learning, which we implement in Section 4.

## 3.2 CAUSAL MODEL IDENTIFICATION

In Section 3.1, we have discussed the compositional properties of hierarchical data-generating processes, where we assume access to the true latent variables $\mathbf{z}$ and $\boldsymbol{\epsilon}$. Here, we establish learning guarantees for recovering the true variables and the hierarchical graph structure (up to certain equivalent classes) from the observed image-text distribution $p(\mathbf{x}, \mathbf{d})$.

We first define identifiability, which formalizes the equivalent class to which our learned representation recovers the true representation.

**Definition 3.2** (Component-wise Identifiability). Let $\mathbf{z} \in \mathcal{Z}$ and $\hat{\mathbf{z}} \in \mathcal{Z}$ be variables under two model specifications $\boldsymbol{\theta}$ and $\hat{\boldsymbol{\theta}}$ respectively. We say that $\mathbf{z}$ and $\hat{\mathbf{z}}$ are *identified component-wise* if there exists a permutation $\pi$ such that for each $i \in [n(\mathbf{z})]$, $\hat{z}_i = h_i(z_{\pi(i)})$ where $h_i$ is invertible.

Here, $\boldsymbol{\theta}$ represents the true model and $\hat{\boldsymbol{\theta}}$ represents the learned version. Under the component-wise identifiability, our learned representation $\hat{z}_i$ captures complete information about a single variable $z_{\pi(i)}$ and no information from other variables $z_j$ with $j \neq \pi(i)$. This notion of identifiability is broadly adopted in the nonlinear independence component analysis literature (ICA) (Hyvarinen & Morioka, 2016; Hyvarinen et al., 2019).

In the following, we introduce and interpret conditions of the data-generating process that lead to component-wise identifiability over all the latent variables $\mathbf{z}$.

**Remarks on the problem and our contribution.** Identifying the latent *hierarchical* models has long been a challenging task. Much previous research has focused on hierarchical models with discrete variables (Pearl, 1988; Zhang, 2004; Choi et al., 2011; Gu & Dunson, 2023; Kong et al., 2024) or assumes linear relations among variables (Xie et al., 2022; Huang et al., 2022; Dong et al., 2024; Anandkumar et al., 2013). Unfortunately, both linearity and discreteness could be too restrictive to model complex real-world distributions of interest in this work (e.g., high-dimensional image distributions). Closely related to our setting is prior work Kong et al. (2023) that admits nonlinear relations among the latent variables. They utilize conditional independence and sparse graphical conditions to provide identifiability guarantees for subspaces of latent variables, where latent dimensions can be entangled within certain groups. While informative in many use cases, such subspace identifiability fails to reflect the granular graphical structure among individual concepts across levels. For instance, multiple concepts at the same level (e.g., "eyes" and "nose") may be mixed into a single subspace, compromising the transferability of these individual concepts and limiting compositionality. In contrast, we utilize the auxiliary information (e.g., the discrete concepts $\mathbf{d}$) and assume that latent variables influence each other in a non-trivial manner, which we formalize in Condition 3.3-iv. These conditions allow us to achieve component-wise identification (as opposed to the subspace identification (Kong et al., 2023)) and fully identify the graphical structure, which is instrumental for compositional generalization.

**Condition 3.3** (Identification Conditions).

> i [Invertibility]: There exists a smooth and invertible map $g_l : (\mathbf{z}_l, \boldsymbol{\epsilon}_l) \mapsto \mathbf{x}$ for $l \in [0, L]$.

*ii [Smooth Density]: The probability density function $p(\mathbf{z}_{l+1}|\mathbf{z}_l)$ is smooth.*

*iii [Conditional Independence]: Components in $\mathbf{z}_{l+1}$ are independent given $\mathbf{z}_l$: $p(\mathbf{z}_{l+1}|\mathbf{z}_l) = \prod_n p(z_{l+1,n}|\mathbf{z}_l)$.*

*iv [Sufficient Variability]: For each value of $\mathbf{z}_{l+1}$, there exist $2n(\mathbf{z}_{l+1}) + 1$ values of $\mathbf{z}_l$, i.e., $\mathbf{z}_l^{(n)}$ with $n = 0, 1, \ldots, 2n(\mathbf{z}_{l+1}) + 1$, such that the $2n(\mathbf{z}_{l+1})$ vectors $\mathbf{w}(\mathbf{z}_{l+1}, \mathbf{z}_l^{(n)}) - \mathbf{w}(\mathbf{z}_{l+1}, \mathbf{z}_l^{(0)})$ are linearly independent, where vector $\mathbf{w}(\mathbf{z}_{l+1}, \mathbf{z}_l)$ is defined as follows:*

$$\mathbf{w}(\mathbf{z}_{l+1}, \mathbf{z}_l) = \left( \frac{\partial \log p(\mathbf{z}_{l+1}|\mathbf{z}_l)}{\partial z_{l+1,1}}, \ldots, \frac{\partial \log p(\mathbf{z}_{l+1}|\mathbf{z}_l)}{\partial z_{l+1,n(\mathbf{z}_{l+1})}}, \frac{\partial^2 \log p(\mathbf{z}_{l+1}|\mathbf{z}_l)}{(\partial z_{l+1,1})^2}, \ldots, \frac{\partial^2 \log p(\mathbf{z}_{l+1}|\mathbf{z}_l)}{\partial(z_{l+1,n(\mathbf{z}_{l+1})})^2} \right).$$

**Discussion and interpretation.** Condition 3.3-i guarantees that the observed variables $\mathbf{x}$ fully preserve the information in $\mathbf{z}$, which is necessary since our goal is to recover $\mathbf{z}$ from $\mathbf{x}$. Intuitively, the information accumulates from top to bottom in the hierarchical model and ultimately manifests as the observed variable $\mathbf{x}$. This is plausible for many applications where $\mathbf{x}$ (e.g., images) can be very high-dimensional and information-rich. This condition is commonly employed in ICA literature (Hyvarinen & Morioka, 2016; Hyvarinen et al., 2019; Khemakhem et al., 2020b;a; von Kügelgen et al., 2021; Kong et al., 2023). Condition 3.3-ii,iii are also standard in the ICA literature. In particular, Condition 3.3-ii is a mild regularity condition on the conditional distributions, allowing us to measure distribution variations with density function derivatives. Condition 3.3-iii assumes that the statistical dependence among latent variables on the same level originates from higher-level variables. For instance, the dependence between a dog's "eye" and "nose" features stems from a higher-level concept like "breed". Condition 3.3-iv formalizes the intuition of "sufficient variation" among the latent variables. In particular, the distributions of distinct low-level concepts $z_{l+1,i}$, $z_{l+1,j}$ *vary differently* in response to their parent variables in $\mathbf{z}_l$. For example, low-level concepts like "eye" and "nose" exhibit different patterns of change when the concept "face" varies, which enables humans to recognize them as separate concepts. This condition is adopted and discussed extensively in prior work (Hyvarinen et al., 2019; Khemakhem et al., 2020a; Kong et al., 2022).

**Theorem 3.4** (Causal Module Identification). *We assume the data-generating process equation 1. Under Condition 3.3, we attain component-wise identifiability of $\mathbf{z}_l$ and the graphical structures $\mathcal{G}$ up to the index permutation at each level $l$.*

**Proof sketch.** The crux is leveraging the influences from the high-level to the low-level latent variables in the hierarchical model. Specifically, we utilize the textual description $\mathbf{d}$ as the initial source of variation to identify the adjacent concepts $\mathbf{z}_1$. With $\mathbf{z}_1$ identified, we can then use these variables to identify its children concepts $\mathbf{z}_2$. This process repeats through each level until we have fully identified all latent variables component-wise with permutation indeterminacy within each level. Classic causal discovery algorithms (e.g., PC algorithm (Spirtes et al., 2000)) can then process these identified latent variables to determine the graphical structure.

**Implications.** Theorem 3.4 guarantees that we can recover the hierarchical data-generating process from the observed distribution $p(\mathbf{d}, \mathbf{x})$. With Theorem 3.1, we demonstrated that if the data generating process of the text-image distribution $p(\mathbf{x}, \mathbf{z})$ satisfies certain favorable properties (e.g., sufficient variability in Condition 3.3-iv), it is theoretically possible to recover this latent process from data. This recovery gives rise to compositional generalization capabilities as shown in Theorem 3.1.

## 4 INTEGRATING HIERARCHICAL STRUCTURES INTO DIFFUSION MODELS

In this section, we integrate key insights from Section 3 into existing diffusion models (Rombach et al., 2022) to enhance compositionality.

**Hierarchical levels and diffusion steps.** We conceptualize a diffusion model as a family of models $\{f_t\}_{t=1}^T$, where each $f_t$ restores $\mathbf{x}_{t+1}$ from its noisier version $\mathbf{x}_t$ by optimizing the variational evidence lower bound objective (Sohl-Dickstein et al., 2015; Ho et al., 2020):

$$\mathcal{L}_d := \sum_{t=1}^{T-1} \mathrm{KL}\left( q(\mathbf{x}_t|\mathbf{x}_{t+1}, \mathbf{x}_0) \,\|\, p_{f_{t+1}}(\mathbf{x}_t|\mathbf{x}_{t+1}, \mathbf{y}) \right) - \log p_{f_1}(\mathbf{x}_0|\mathbf{x}_1, \mathbf{y}), \tag{2}$$

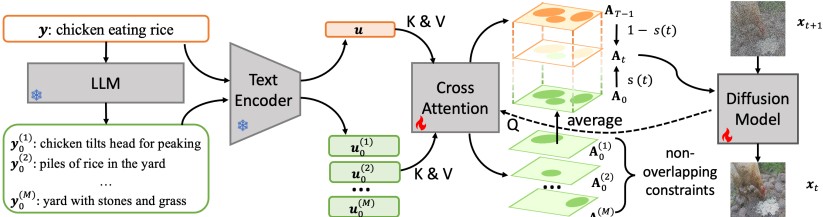

Figure 4: **HierDiff**. We first generate low-level text descriptions $\{\mathbf{y}_0^{(m)}\}_{m=1}^M$ from the original global description $\mathbf{y}$ and obtain their embeddings $\mathbf{u}$ and $\{\mathbf{u}_0^{(m)}\}_{m=1}^M$. We average the $M$ low-level cross-attention outputs $\{\mathbf{A}_0^{(m)}\}_{m=1}^M$ and interpolate it with the global cross-attention-map $\mathbf{A}_{T-1}$ according to a step-dependent function $s(t)$. The resultant $\mathbf{A}_t$ smoothly transitions from $\mathbf{A}_{T-1}$ to $\mathbf{A}_0$ in the generating process. We impose non-overlapping constraints to minimize unnecessary interactions among low-level concepts.

where $q(\mathbf{x}_t|\mathbf{x}_{t+1}, \mathbf{x}_0)$ denotes the reverse diffusion process, KL stands for KL divergence, and $\mathbf{y}$ refers to conditioning information (e.g., text). As interpreted in prior work (Kong et al., 2024), $f_{t+1}$ extracts representation $\mathbf{z}_{\mathcal{S}(t+1)}$ from the noisy data $\mathbf{x}_{t+1}$, and then employs $\mathbf{z}_{\mathcal{S}(t+1)}$ and additional information $\mathbf{y}$ to recover $\mathbf{x}_t$. Here, $\mathbf{z}_{\mathcal{S}(t)}$ denotes latent variables with indices in a $t$-dependent set $\mathcal{S}(t)$. Higher noise levels (large $t$) corrupt low-level concepts in $\mathbf{x}_t$, so the representation $\mathbf{z}_{\mathcal{S}(t)}$ only retains high-level concepts. Therefore, a higher noise level (a larger $t$) corresponds monotonically to a higher concept level $\mathcal{S}(t)$. In Figure 3, if noise level $t$ just suffices to obscure low-level concepts $\mathbf{z}_2$ (e.g., "beak"), then $\mathbf{z}_{\mathcal{S}(t)}$ corresponds to high-level concepts $\mathbf{z}_1$ (e.g., "peacock").

**Hierarchical concept injection with sparsity control.** Conventional approaches condition all generation steps with a global text prompt $\mathbf{y}$ (Rombach et al., 2022). However, our hierarchical-level interpretation suggests that only the information gap between $\mathbf{z}_{S(t+1)}$ and $\mathbf{z}_{S(t)}$ is needed at step $t$. Applying a global, invariant conditioning $\mathbf{y}$ can limit the model's capacity, since it is compelled to disentangle and extract the desired information at each step. Moreover, this approach overlooks the naturally sparse structures in the hierarchical model, which may create unnecessary concept interactions and compromise composability as discussed in Section 3. Based on these insights, we formulate two key goals to improve existing methods:

1. **Goal 1**: Inject step-specific information $\mathbf{y}_t$;
2. **Goal 2**: Encourage sparse interactions among concepts.

For **Goal 1**, we produce detailed textual descriptions $\mathbf{y}_0 := \{\mathbf{y}_0^{(m)}\}_{m=1}^M$ for $M$ low-level concepts from the original high-level textual description $\mathbf{y}$. This can be accomplished via a language model, as shown in prior work Feng et al. (2024); Lian et al. (2023); Wu et al. (2023); Yang et al. (2024b) (see more in Appendix C.2). We treat these low-level descriptions as the information to be injected at the final step from $\mathbf{x}_1$ to $\mathbf{x}_0$ and the high-level description $\mathbf{y}$ as information at the initial step from $\mathbf{x}_T$ to $\mathbf{x}_{T-1}$. For the intermediate steps $0 < t < T-1$, we interpolate the cross-attention outputs between the global text $\mathbf{y}$ and $\mathbf{x}_t$, denoted as $\mathbf{A}_{T-1} := \mathrm{XAttn}(\mathbf{x}_t, \mathbf{y})$, and cross-attention outputs between low-level descriptions $\{\mathbf{y}_0^{(m)}\}_{m=1}^M$ and $\mathbf{x}_t$, denoted as $\mathbf{A}_0^{(m)} := \mathrm{XAttn}(\mathbf{x}_t, \mathbf{y}_0^{(m)})$:

$$\mathbf{A}_t := (1 - s(t)) \cdot \mathbf{A}_{T-1} + \frac{s(t)}{M} \cdot \sum_{m=1}^M \mathbf{A}_0^{(m)}, \tag{3}$$

where $s(t)$ is a monotonically decreasing function with $s(0) = 1$ and $s(T-1) = 0$. We apply this modified cross-attention $\mathbf{A}_t$ at step $t$ for conditioning. In this manner, we control the granularity of the injected information to match the diffusion step (i.e., high-level concepts at large steps).

For **Goal 2**, we impose sparse regularization $\mathcal{L}_n$ on the overlaps among cross-attention maps $\{\mathbf{H}_0^{(m)}\}_{m=1}^M$ from low-level descriptions $\{\mathbf{y}_0^{(m)}\}_{m=1}^M$ to minimize unnecessary spatial interactions among the $M$ low-level concepts:

$$\mathcal{L}_n := \sum_{m,n \in [M]: m \neq n} D\left(\mathbf{H}_0^{(m)}, \mathbf{H}_0^{(n)}\right), \tag{4}$$

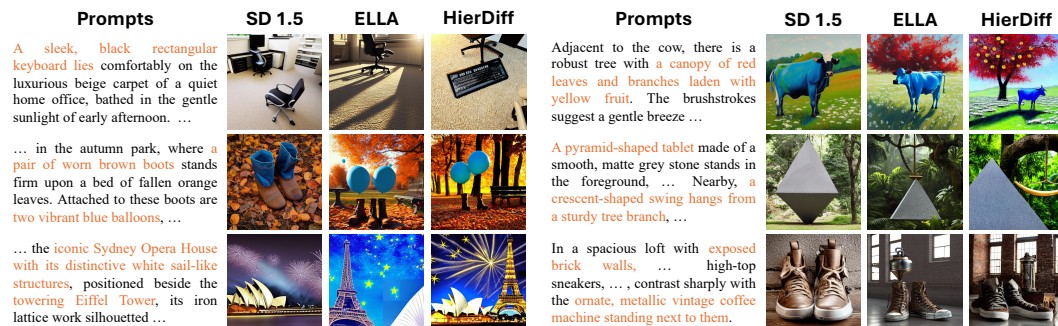

Figure 5: **Text-to-image generation results.** We highlight distinguishing tokens.

Table 1: **Evaluation results on DPG-Bench (Hu et al., 2024)**. All baseline results are obtained from Hu et al. (2024).

| Model | Average | Global | Entity | Attribute | Relation | Other |
|---|---|---|---|---|---|---|
| SD v2 Rombach et al. (2022) | 68.09 | 77.67 | 78.13 | 74.91 | 80.72 | 80.66 |
| PixArt-$\alpha$ Chen et al. (2023) | 71.11 | 74.97 | 79.32 | 78.60 | 82.57 | 76.96 |
| Playground v2 Li et al. (2023a) | 74.54 | 83.61 | 79.91 | 82.67 | 80.62 | 81.22 |
| SD v1.5 Rombach et al. (2022) | 63.18 | 74.63 | 74.23 | 75.39 | 73.49 | 67.81 |
| ELLA Hu et al. (2024) | 74.91 | 84.03 | 84.61 | 83.48 | 84.03 | 80.79 |
| **HierDiff** | **79.56** | **87.81** | **86.37** | **86.86** | **87.76** | **84.41** |

where the DICE loss (Sudre et al., 2017; Yeung et al., 2023) $D(\mathbf{H}_1, \mathbf{H}_1) := \frac{2 \cdot \mathrm{tr}(\mathbf{H}_1 \mathbf{H}_2)}{\|\mathbf{H}_1\|_1 + \|\mathbf{H}_2\|_1}$ measures the spatial overlap between attention maps $\mathbf{H}_1$ and $\mathbf{H}_2$. Under this regularization, concepts overlap sparsely with each other at each level, promoting sparse connectivity and thus composability.

The overall training objective becomes:

$$\mathcal{L} := \mathcal{L}_{\mathrm{d}} + \lambda \cdot \mathcal{L}_{\mathrm{n}}, \tag{5}$$

where $\lambda$ controls the regularization $\mathcal{L}_{\mathrm{n}}$. We refer to our method as **HierDiff** (Figure 4).

**Theory & practice.** Our theoretical conditions and implementation are related as follows: 1) *Hierarchical processes* (Eq. 1 & Cond. 3.3-iii): The iterative diffusion chain naturally respects this hierarchical structure. 2) Sparse connectivity (Thm. 3.1): We enforce this key condition by minimizing attention overlap, an effective and practical surrogate as validated in our ablations. 3) *Level-dependent transformations* (Eq. 1): Time-indexed diffusion models provide the required flexible, level-dependent transformations. 4) Invertibility (Cond. 3.3-i): The model's reconstruction objective inherently promotes invertibility. While other conditions (Condition 3.3-ii,iv) are assumptions on the data distribution itself. Although directly verifying the latent graph structure on real-world data is presently challenging, our work provides a clear mapping from abstract theoretical principles to concrete implementation choices. The strong empirical performance of **HierDiff**, as we will show in Section 5, validates the utility of our framework. Thus, this foundational understanding serves as a roadmap for future progress in compositional generalization.

## 5 EXPERIMENTS

**Setup.** We fine-tune **HierDiff** from Stable Diffusion v1.5 Rombach et al. (2022). Following Hu et al. (2024), we replace the CLIP text encoder with FLAN-T5-xl (Raffel et al., 2020) for enhanced text understanding capabilities, which we freeze during training. For training, we use the public LayoutSAM dataset Zhang et al. (2024a), which contains both the high-level text $\mathbf{y}$ and corresponding low-level, local descriptions $\mathbf{y}_0^{(m)}$. At test time, given text $\mathbf{y}$, we apply QWEN-v2.5 Yang et al. (2024a) to generate low-level local text descriptions similar to the training dataset (details in App. C.2). We adopt the interpolation function $s(t) = \cos\left(\frac{\pi \cdot t}{2(T-1)}\right)$, the number of local concepts $M = 3$, and weight $\lambda = 1e-4$ throughout our experiments. We adopt DPG-Bench (Hu et al., 2024), which introduces five metrics, namely, Global, Entity, Attributes, Relation, and Other. The dataset comprises $1,065$ text prompts, each involving multiple objects/concepts with various relations.

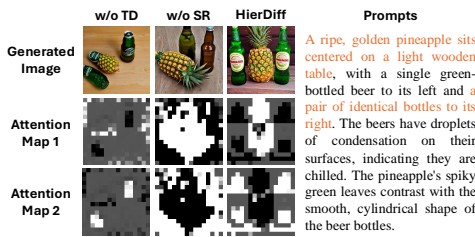

Figure 6: **Ablation studies.**

Table 2: **Ablation studies on the DPG-Bench Hu et al. (2024)**. TD and SR stand for time dependence and sparsity regularization, respectively.

| Metrics | -w/o TD | -w/o SR | **HierDiff** |
|---|---|---|---|
| Global | 88.84 | 84.30 ↓ | 87.81 |
| Entity | 85.73 ↓ | 85.52 ↓ | 86.37 |
| Attribute | 85.46 ↓ | 87.14 | 86.86 |
| Relation | 84.83 ↓ | 87.03 ↓ | 87.76 |
| Other | 85.16 | 83.48 ↓ | 84.41 |

**Comparison with baselines.** In Table 1, **HierDiff** outperforms baseline methods across all evaluation metrics, demonstrating its superior capability in handling complex prompts involving multiple concepts and relationships. Figure 5 visualizes the results (more in Appendix C.3). For the first prompt, only **HierDiff** successfully captures the "keyboard" concept and correctly renders its attributes (e.g., "black", "sleek"), while baselines completely overlook this concept.

**Ablation studies.** We conduct ablation studies by sequentially removing the sparsity regularization $\mathcal{L}_n$ and then the time dependence (i.e., using only the global text $\mathbf{y}$). Table 2 shows that both components contribute positively to the overall performance. Notably, the time dependence significantly aids the model to understand complex relations among concepts ("Relation" from 84.83 to 83.03), demonstrating the benefits of hierarchical structure to organize multiple concepts. The sparsity regularization allows for precise control of individual concepts, universally improving all metrics. Figure 6 visually demonstrates these findings. Without sparsity constraints, the model's attention map lumps the two bottles together. Comparing the attention maps from two local prompts, we observe that the sparsity constraint reduces the overlapping areas, enabling the model to control concepts separately. Without the time dependence, the model fails to capture the relation between concepts, resulting in a confused mixture of "pineapple" and "beer". See more examples in Appendix C.4.

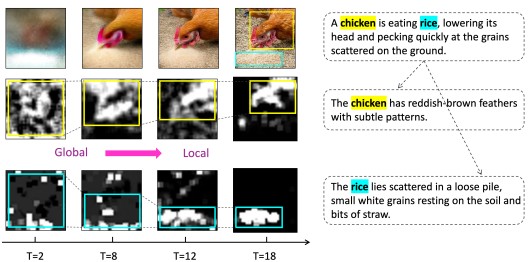

Figure 7: **Local cross-attention maps over steps.**

Table 3: **Comparison with SOTA T2I models.**

| Method | DPG ↑ |
|---|---|
| DALLE 3 (OpenAI, 2023) | 83.5 |
| SD3-medium (Esser et al., 2024) | 84.1 |
| FLUX-dev (B. F. Labs, 2024) | 84.0 |
| FLUX-schnell (B. F. Labs, 2024) | 84.8 |
| SANA-1.0 (Xie et al., 2024) | 83.6 |
| SANA-1.5 (Xie et al., 2025) | 84.7 |
| **HierDiff**-DiT | **84.9** |

**Scalability.** To validate the scalability of our approach, we extend the implementations from the U-Net architecture to diffusion transformers with 4.8B parameters. As shown in Table 3, after scaling, we can achieve comparable performance with billion-scale models. Details in Appendix C.1.

**Visualization of composition.** Figure 7 visualizes the cross-attention maps from two low-level text descriptions ("chicken" and "rice") across the diffusion steps with our model. We can observe that under the two local-level text descriptions gradually shift from dispersed global attentions to more focused local attentions, and the intersection remains minimal. This verifies that our method can indeed facilitate composition by proper decomposition and re-composition with minimal interference.

## 6 CONCLUSION AND LIMITATIONS

In this work, we connect compositional generalization with humans' cognitive process of drawing analogies. We formalize this process via a hierarchical latent model that embodies causal modularity and minimal-change principles. Our framework accommodates complex concept interactions without restrictive assumptions. These theory insights lead to **HierDiff**, a T2I model that possesses competitive composition capabilities. **Limitations.** Condition 3.3-iii assumes no direct causal influences among variables on each hierarchical level, which may restrict the representative power. One may mitigate this with additional hierarchical levels to convert within-level to cross-level influences, or alternatively, consider more involved conditions in recent work (Zhang et al., 2024b).

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

# Appendix

**The use of large language models (LLMs)**. We employ LLMs to locate typos and polish certain text in the paper. LLMs play no part in the idealization.

## A  RELATED WORK

**Compositional generalization.** Compositional generalization has garnered significant attention from the generative model community, especially for text-to-image generation. One avenue explores fine-tuning text-to-image models by incorporating feedback from image understanding systems as a form of reward (Huang et al., 2024; Xu et al., 2024; Sun et al., 2023; Fang et al., 2023). However, this strategy may be limited by the text comprehension capabilities of models like CLIP. Another approach involves adjusting the models' cross-attention mechanisms to align with the spatial and semantic details specified in the prompts (Liu et al., 2022; Bar-Tal et al., 2023; Li et al., 2023b; Rassin et al., 2024; Chefer et al., 2023; Feng et al., 2023a; Chen et al., 2024; Xie et al., 2023; Kim et al., 2023). This approach relies on the interpretability of the foundational models and often results in only broad, suboptimal control over the generated images. By leveraging the planning and reasoning strengths of language models, researchers have also broken down complex prompts into multiple regional descriptions, providing more precise conditions to guide the image generation process (Cho et al., 2023; Feng et al., 2023b; Wang et al., 2024; Yang et al., 2024b; Lian et al., 2023; Feng et al., 2024). This decomposition aids in creating images that more accurately reflect the detailed components of the prompts. These methods operate at the inference time and do not fundamentally learn disentangled concepts. Recent work (Hu et al., 2024; Wu et al., 2023) utilizes diffusion timesteps to modify the text embedding for refined generation control. Nevertheless, Hu et al. (2024) do not consider the spatial relations among concepts and fully rely on the pre-trained diffusion model's capacity. Wu et al. (2023) introduce additional inference-time optimization overhead and depend on the CLIP score as the optimization objective, which limits the generation quality with CLIP's capacity. Guided by our theoretical insights, our work imposes proper constraints and modifications on the cross-entropy to learn disentangled concepts and their relations.

Although empirical studies are abundant in the field (Du & Kaelbling, 2024; Liu et al., 2022; Zhang et al., 2024c; Hu et al., 2024; Huang et al., 2023; Bar-Tal et al., 2023; Yang et al., 2024b), theoretical understanding remains limited and often hinges on restrictive assumptions about concept interactions. Specifically, Brady et al. (2023); Wiedemer et al. (2024a) consider concepts that affect disjoint pixel regions, effectively eliminating interaction between them. Lachapelle et al. (2023) models the influences of concepts on the pixel space as purely additive, an approach that Brady et al. (2024) extends to include second-order polynomial terms. Additionally, Wiedemer et al. (2024b) assumes direct access to the function governing concept interactions. These theoretical works also tend to overlook the varying levels of abstraction among concepts and their relationships within the latent space. In contrast, thanks to the hierarchical structure, our theory admits compositions of transformations across hierarchical levels, allowing for complex interaction among concepts at different hierarchical levels.

**Latent hierarchical model identification.** Modeling complex real-world data requires capturing hierarchical structures among latent variables. Prior work has explored identification conditions for such hierarchies with continuous latent variables influencing each other linearly (Xie et al., 2022; Huang et al., 2022; Dong et al., 2024; Anandkumar et al., 2013). Other studies focus on fully discrete cases, limiting their applicability to continuous data like images (Pearl, 1988; Zhang, 2004; Choi et al., 2011; Gu & Dunson, 2023). Moreover, latent tree models connect variables through a single undirected path (Pearl, 1988; Zhang, 2004; Choi et al., 2011), which may oversimplify complex relationships. Closely related to ours, Kong et al. (2023) address nonlinear, continuous latent hierarchical models. However, their framework cannot identify latent variables component-wise, leaving room for concept entanglement. In contrast, we provide component-wise identifiability for latent variables and the graphical structures, along with transparent conditions for the data-generating process.

# B  PROOFS FOR THEORETICAL RESULTS

## B.1  PROOF FOR THEOREM 3.1

**Theorem 3.1** (Composition Generalization). *We assume the data-generating process equation 1. The discrete concept combination $\mathbf{d}$ is composable (i.e., $\mathbf{d} \in \Omega_{\mathrm{comp}}$) if for each continuous latent variable $z \in \mathbf{z}$, its parents' distribution support $\mathrm{supp}(\mathrm{Pa}(z)|\mathbf{d})$ is contained by $\mathrm{supp}(\mathrm{Pa}(z)|\tilde{\mathbf{d}})$ for some combination $\tilde{\mathbf{d}} \in \Omega_{\mathrm{supp}}$ on the support, i.e., $\mathrm{supp}(\mathrm{Pa}(z)|\mathbf{d}) \subseteq \mathrm{supp}(\mathrm{Pa}(z)|\tilde{\mathbf{d}})$.*

*Proof.* By definition, the concept combination $\mathbf{d}$ is composable (i.e., $\mathbf{d} \in \Omega_{\mathrm{comp}}$) when the two alternative model specifications $\boldsymbol{\theta}$ and $\hat{\boldsymbol{\theta}}$ agree on this specific $\mathbf{d}$, i.e., $\hat{g}_z = g_z$ for any $z \in \mathbf{z}$ over its inputs' support $\mathcal{S}_z(\mathbf{d}) := \mathrm{supp}(\mathrm{Pa}(z)|\mathbf{d}) \times \mathrm{supp}(\boldsymbol{\epsilon}_z)$. We note that each exogenous variable $\epsilon_z$ is independent of $\mathrm{Pa}(z)$ and its distribution remains invariant to the discrete variable $\mathbf{d}$. We denote this relation as $\boldsymbol{\theta}|_{\mathbf{d}} = \hat{\boldsymbol{\theta}}|_{\mathbf{d}}$.

To derive this relation, we first show that under the assumption of the hierarchical data-generating process equation 1, the specific model $\boldsymbol{\theta} := \big(p(\mathbf{z}_1, \mathbf{d}), \{g_v\}_{v \in \mathcal{V} \setminus (\mathbf{z}_1, \mathbf{d})}\big)$'s behavior on the discrete concept space $\Omega_{\mathrm{comp}}$ is fully determined by its behavior on the support $\Omega_{\mathrm{supp}}$. That is, if two specifications $\boldsymbol{\theta}$ and $\hat{\boldsymbol{\theta}}$ follow the hierarchical model assumption equation 1 and their behavior match over the support $\Omega_{\mathrm{supp}}$, this agreement would extend to $\Omega_{\mathrm{comp}}$: $\forall \tilde{\mathbf{d}} \in \Omega_{\mathrm{supp}}, \boldsymbol{\theta}|_{\tilde{\mathbf{d}}} = \hat{\boldsymbol{\theta}}|_{\tilde{\mathbf{d}}} \implies \forall \mathbf{d} \in \Omega_{\mathrm{comp}}, \boldsymbol{\theta}|_{\mathbf{d}} = \hat{\boldsymbol{\theta}}|_{\mathbf{d}}$.

To this end, we assess the elementary generating function $z := g_z(\mathrm{Pa}(z), \epsilon_z)$ for every $z \in \mathbf{z}$ present in the hierarchical model. Although latent variables $\{\mathbf{z}_l\}_{l \in [L+1]}$ form a Markov chain, the first module $p(\mathbf{z}_1|\mathbf{d})$ may yield distinct supports $\mathrm{supp}(\mathbf{z}_1|\mathbf{d})$ across various values of $\mathbf{d}$ (e.g., $d = 0$ for absence of the concept). Consequently, the matching of two models $\boldsymbol{\theta}$ and $\hat{\boldsymbol{\theta}}$ is only partially supported and depends on the specific value of $\mathbf{d}$. We characterize a potentially larger composable space $\Omega_{\mathrm{comp}}$ given their matching over the training support $\Omega_{\mathrm{supp}}$. Under the theorem condition, we have $\mathrm{supp}(\mathrm{Pa}(z)|\mathbf{d})$ at the specific $\mathbf{d}$ is fully contained by $\mathrm{supp}(\mathrm{Pa}(z)|\tilde{\mathbf{d}}(\mathbf{d}))$ at some $\tilde{\mathbf{d}}(\mathbf{d}) \in \Omega_{\mathrm{supp}}$ dependent on $\mathbf{d}$, i.e.,

$$\mathrm{supp}(\mathrm{Pa}(z)|\mathbf{d}) \subseteq \mathrm{supp}(\mathrm{Pa}(z)|\tilde{\mathbf{d}}(\mathbf{d})). \tag{6}$$

As the two models $g_z$ and $\hat{g}_z$ match over the discrete support $\Omega_{\mathrm{supp}}$, this equality relation in equation 6 implies that this equality extends to $\tilde{\mathbf{d}}(\mathbf{d})$:

$$g_z = \hat{g}_z, \forall (\mathrm{Pa}(z), \epsilon_z) \in \mathcal{S}_z(\tilde{\mathbf{d}}(\mathbf{d})). \tag{7}$$

As the relation in equation 7 holds true for all modules of $\boldsymbol{\theta}$ and $\hat{\boldsymbol{\theta}}$, the equality extends to the entire hierarchical model, i.e., $\boldsymbol{\theta}|_{\mathbf{d}} = \hat{\boldsymbol{\theta}}_{\mathbf{d}}$ for $\mathbf{d} \in \Omega_{\mathrm{comp}}$, which concludes our proof.

$\square$

## B.2  PROOF FOR THEOREM 3.4

**Condition 3.3** (Identification Conditions).

   *i [Invertibility]: There exists a smooth and invertible map $g_l : (\mathbf{z}_l, \epsilon_l) \mapsto \mathbf{x}$ for $l \in [0, L]$.*

   *ii [Smooth Density]: The probability density function $p(\mathbf{z}_{l+1}|\mathbf{z}_l)$ is smooth.*

   *iii [Conditional Independence]: Components in $\mathbf{z}_{l+1}$ are independent given $\mathbf{z}_l$: $p(\mathbf{z}_{l+1}|\mathbf{z}_l) = \prod_n p(z_{l+1,n}|\mathbf{z}_l)$.*

   *iv [Sufficient Variability]: For each value of $\mathbf{z}_{l+1}$, there exist $2n(\mathbf{z}_{l+1}) + 1$ values of $\mathbf{z}_l$, i.e., $\mathbf{z}_l^{(n)}$ with $n = 0, 1, \ldots, 2n(\mathbf{z}_{l+1}) + 1$, such that the $2n(\mathbf{z}_{l+1})$ vectors $\mathbf{w}(\mathbf{z}_{l+1}, \mathbf{z}_l^{(n)}) - \mathbf{w}(\mathbf{z}_{l+1}, \mathbf{z}_l^{(0)})$ are linearly independent, where vector $\mathbf{w}(\mathbf{z}_{l+1}, \mathbf{z}_l)$ is defined as follows:*

$$\mathbf{w}(\mathbf{z}_{l+1}, \mathbf{z}_l) = \Big( \frac{\partial \log p(\mathbf{z}_{l+1}|\mathbf{z}_l)}{\partial z_{l+1,1}}, \ldots, \frac{\partial \log p(\mathbf{z}_{l+1}|\mathbf{z}_l)}{\partial z_{l+1,n(\mathbf{z}_{l+1})}}, \frac{\partial^2 \log p(\mathbf{z}_{l+1}|\mathbf{z}_l)}{(\partial z_{l+1,1})^2}, \ldots, \frac{\partial^2 \log p(\mathbf{z}_{l+1}|\mathbf{z}_l)}{\partial (z_{l+1,n(\mathbf{z}_{l+1})})^2} \Big).$$

**Theorem B.1** (Causal Module Identification). *We assume the data-generating process equation 1. Under Condition 3.3, we attain component-wise identifiability of $\mathbf{z}_l$ and the graphical structures $\mathcal{G}$ up to the index permutation at each level $l$.*

*Proof.* We introduce Lemma B.1 from Kong et al. (2022), which identifies a trivial hierarchical model with only one latent level, i.e., $L = 1$.

**Lemma B.1** (Single-level Identification (Kong et al., 2022)). *We assume the following data-generating process equation 1:*

$$\mathbf{z} \sim p(\mathbf{z}|\mathbf{u}), \quad \boldsymbol{\epsilon} \sim p(\boldsymbol{\epsilon}), \quad \mathbf{x} := g(\mathbf{z}, \boldsymbol{\epsilon}), \tag{8}$$

*where $\boldsymbol{\epsilon}$ refers to the exogenous variable independent of $\mathbf{z}$ and $g$ stands for the generating function. Under Condition 3.3 with $L = 1$ and $\mathbf{z}_0 = \mathbf{u}$, we attain component-wise identifiability of $\mathbf{z}_1$.*

In the general hierarchical case, we view the observed discrete variable $\mathbf{d}$ as the top-level variable $\mathbf{u}$ in equation 8 as the starting point. Lemma B.1 implies the component-wise identifiability of $\mathbf{z}_1$. We then iteratively apply Lemma B.1 to identify level $\mathbf{z}_{l+1}$ sequentially from top to bottom equation 1 by viewing the previously identified level $\mathbf{z}_l$ as the conditioning variable $\mathbf{u}$ in equation 8. This reasoning gives the component-wise identifiability results for the entire hierarchical model.

Since all the latent variables $\{z_i\}_{i=1}^{n(\mathbf{z})}$, we can view them as the observed variables. The identifiability of the graphical structure $\mathcal{G}$ follows from classic causal discovery methods (i.e., PC algorithm (Spirtes et al., 2000)).

$\square$

# C ADDITIONAL DETAILS FOR EXPERIMENTS

## C.1 SETUP DETAILS

We train the model with a batch size of $800$ and a learning rate of $5e - 5$. To inject multiple text conditions, we replicate the key and value linear layers in cross-attention, inspired by IP-Adapter Ye et al. (2023). During testing, we prompt QWEN2.5 (Yang et al., 2024a) with the instruction "given a prompt X, segment it into three non-overlap descriptions (i.e., any two descriptions are not describing the same object), rewrite each subcaption to avoid interactions across each subcaption." For the experiments in Table 3, we employ the LayoutSAM dataset (Zhang et al., 2024a) and finetune SANA-1.5 (Xie et al., 2025) with a batch size of $576$ for $20000$ steps at a learning rate of $5e - 5$. We choose $\lambda$ from $\{0.1, 1\}$.

## C.2 LANGUAGE MODEL USAGE

We follow established practices Feng et al. (2024); Lian et al. (2023); Wu et al. (2023); Yang et al. (2024b) to instruct QWENv2.5 (Yang et al., 2024a) with a fixed instruction. For example, QWENv2.5 rewrites "a peacock is eating ice cream while..." into "A peacock is in the act of eating", ..., "a serving of ice cream is being visibly diminished". In our evaluation, we've found that QWEN2.5 performs decently for most examples, and more advanced models (Gemini 2.5 Flash, Claude 4) are superior on rare, challenging examples involving dense interactions of multiple concepts (e.g., detailed description of multiple mutually overlapping clothing items on a person). To quantify the performance of QWEN2.5, we instruct Claude 4 to evaluate the presence of high-level concepts in captions processed by QWEN2.5 and observe a $96\%$ success rate over $100$ DPG evaluation prompts. We believe that the advancement of language models could further improve the performance.

## C.3 ADDITIONAL SAMPLES FOR FIGURE 5

Figure 8 and Figure 9 display generated examples from **HierDiff** and baselines, with full text prompts.

## C.4 Additional Samples for Figure 6

Figure 10 displays more examples for the ablation experiments in Figure 6.

## C.5 More Empirical Understanding

While implicit models can be highly expressive, they can struggle with compositional generalization as many solutions might fit the training data but not generalize beyond. Our work introduces a theoretically motivated sparsity constraint (Eq. 5) to select more generalizable solutions. Following your suggestion, we've added fine-grained qualitative analysis in Fig. 11. In Fig. 11(a), our model attends to "cat" (L1) and "sunglasses" (L2) separately, and the baseline attends to all regions and omits "sunglasses". Similarly, in Fig. 11(b), our model, with sparse constraints focusing attention (L1 on "bear" and L2 on "cat"), renders both; the baseline's simultaneous generation misses "cat". The analysis also highlights cases challenging to our model, such as the difficulty in decomposing words and printing the resultant letters correctly (L1 at 901 covers all letters simultaneously) in Fig. 11(c). While extreme sparsity can affect performance in dense interaction scenes (e.g., missing "herb" in Fig. 11(d)), the model's superior performance over the baseline on examples here and all benchmarks confirms its robustness for these scenarios.

| **Prompt** | SD1.5 Rombach et al. (2022) et al. (2024) | ELLA Hu **Ours** |
|---|---|---|

A sleek, black rectangular keyboard lies comfortably on the luxurious beige carpet of a quiet home office, bathed in the gentle sunlight of early afternoon. The keys of the keyboard show signs of frequent use, and it's positioned diagonally across the plush carpet, which is textured with subtle patterns. Nearby, a rolling office chair with a high back and adjustable armrests sits invitingly, hinting at a quick break taken by its usual occupant.

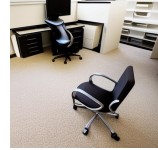 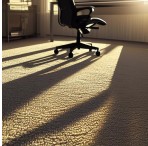 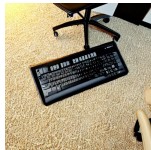

In the fading light of late afternoon, a scene unfolds in the autumn park, where a pair of worn brown boots stands firm upon a bed of fallen orange leaves. Attached to these boots are two vibrant blue balloons, gently swaying in the cool breeze. The balloons cast soft shadows on the ground, nestled among the trees with their leaves transitioning to auburn hues. Nearby, a wooden bench sits empty, inviting passersby to witness the quiet juxtaposition of the still footwear and the dancing balloons.

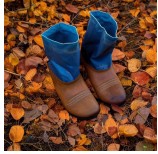 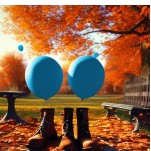 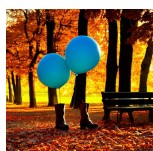

A surreal composite image showcasing the iconic Sydney Opera House with its distinctive white sail-like structures, positioned improbably beside the towering Eiffel Tower, its iron lattice work silhouetted against the night. The backdrop is a vibrant blue sky, pulsating with dynamic energy, where yellow stars burst forth in a dazzling display, and swirls of deeper blue spiral outward. The scene is bathed in an ethereal light that highlights the contrasting textures of the smooth, shell-like tiles of the Opera House and the intricate metalwork of the Eiffel Tower.

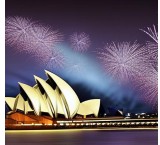 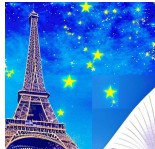 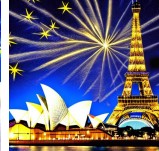

An impressionistic painting depicts a vibrant blue cow standing serenely in a field of delicate white flowers. Adjacent to the cow, there is a robust tree with a canopy of red leaves and branches laden with yellow fruit. The brushstrokes suggest a gentle breeze moving through the scene, and the cow's shadow is cast softly on the green grass beneath it.

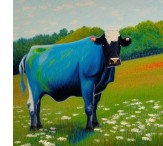 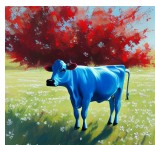 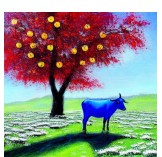

a pyramid-shaped tablet made of a smooth, matte grey stone stands in the foreground, its sharp edges contrasting with the wild, verdant foliage of the surrounding jungle. nearby, a crescent-shaped swing hangs from a sturdy tree branch, crafted from a polished golden wood that glimmers slightly under the dappled sunlight filtering through the dense canopy above. the swing's smooth surface and gentle curve invite a sense of calm amidst the lush greenery.

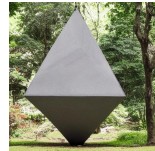 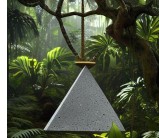 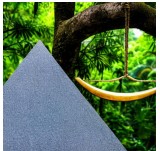

In a spacious loft with high ceilings and exposed brick walls, the morning light filters through large windows, casting a soft glow on a pair of trendy, high-top sneakers. These sneakers, made of rugged leather with bold laces, contrast sharply with the ornate, metallic vintage coffee machine standing next to them. The coffee machine, with its intricate details and polished finish, reflects the light beautifully, setting a striking juxtaposition against the practical, street-style footwear on the polished concrete floor.

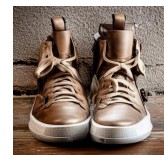 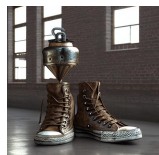 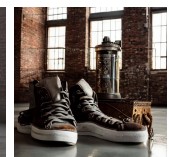

| **Prompt** | SD1.5 Rombach et al. (2022) et al. (2024) | ELLA Hu **Ours** |
|---|---|---|

An elegant pair of glasses with a unique, gold hexagonal frame laying on a smooth, dark wooden surface. The thin metal glints in the ambient light, highlighting the craftsmanship of the frame. The clear lenses reflect a faint image of the room's ceiling lights. To the side of the glasses, a leather-bound book is partially open, its pages untouched.

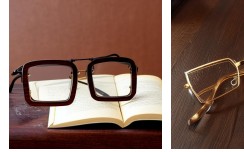 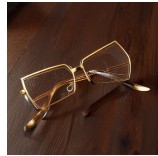 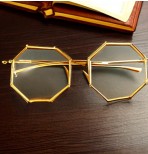

Two multicolored butterflies with delicate, veined wings gently balance atop a vibrant, orange tangerine in a bustling garden. The tangerine, with its glossy, dimpled texture, is situated on a wooden table, contrasting with the greenery of the surrounding foliage and flowers. The butterflies, appearing nearly small in comparison, add a touch of grace to the scene, complementing the natural colors of the verdant backdrop.

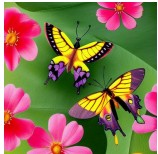 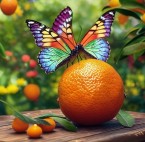 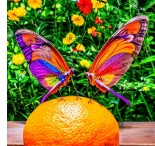

Two sleek blue showerheads, mounted against a backdrop of white ceramic tiles, release a steady stream of water. The water cascades down onto OR-ANGEa vivid, crisp green pearthat is centrally positioned directly beneath them. The pear's smooth and shiny surface gleams as the water droplets rhythmically bounce off, creating a tranquil, almost rhythmic sound in the otherwise silent bathroom.

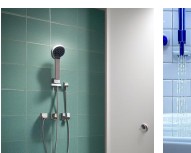 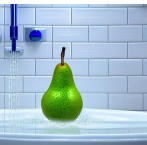 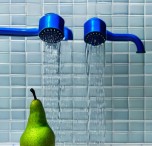

In a modern kitchen, a square, chrome toaster with a sleek finish sits prominently on the marble countertop, its size dwarfing the nearby red vintage rotary telephone, which is placed quaintly on a wooden dining table. The telephone's vibrant red hue contrasts with the neutral tones of the kitchen, and its cord coils gracefully beside it. The polished surfaces of both the toaster and the telephone catch the ambient light, adding a subtle shine to their respective textures.

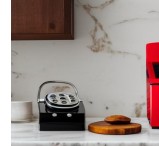 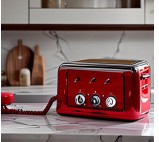 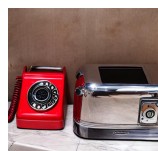

Two slender bamboo-colored chopsticks lie diagonally atop a smooth, round wooden cutting board with a rich grain pattern. The chopsticks, tapered to fine points, create a striking contrast against the cutting board's more robust and circular form. Around the board, there are flecks of freshly chopped green herbs and a small pile of julienned carrots, adding a touch of color to the scene.

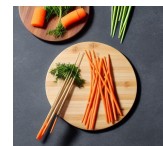 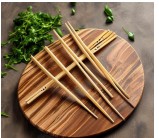 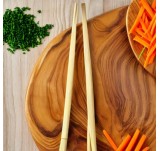

A cozy bathroom features a pristine, white clawfoot bathtub on a backdrop of pastel green tiles. Adjacent to the tub, a tower of soft, white toilet paper is neatly stacked, glimmering gently in the diffuse glow of the afternoon sunlight streaming through a frosted window. The gentle curvature of the tub contrasts with the straight lines of the stack, creating a harmonious balance of shapes within the intimate space.

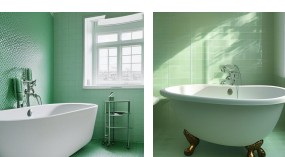 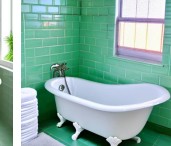

Figure 9: **More text-to-image generation results.**

| | Prompt | w/o TD | w/o SR | **HierDiff** |
|---|---|---|---|---|

A vibrant pink pig trots through a snowy landscape, a bright blue backpack strapped securely to its back. The pig's thick coat contrasts with the soft white blanket of snow that covers the ground around it. As it moves, the blue backpack stands out against the pig's colorful hide and the winter scene, creating a striking visual amidst the serene, frost-covered backdrop.

An outsized dolphin with a sleek, gray body glides through the blue waters, while a small, fluffy chicken with speckled brown and white feathers stands on the nearby sandy shore, appearing diminutive in comparison. The dolphin's fins cut through the water, creating gentle ripples, while the chicken pecks at the ground, seemingly oblivious to the vast size difference. The stark contrast between the dolphin's smooth, aquatic grace and the chicken's terrestrial, feathered form is highlighted by their proximity to one another.

Figure 10: **More ablation studies.** Without time-dependence (TD), the model fails to understand the relationship among the objects in the prompt. Without sparsity regularization (SR), the influence of each prompt could be large, e.g., the attention map of local prompt 1 covers the pineapple and beers. Combining the two proposed designs, **HierDiff** generates images that accurately follow the complex text prompt.

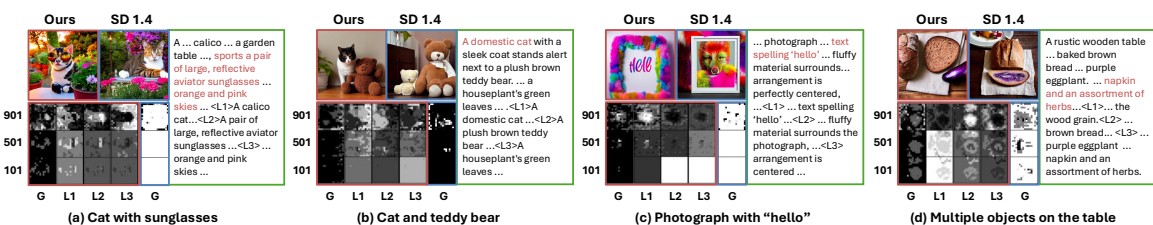

Figure 11: **Fine-grained comparison between our method and Stable Diffusion 1.4.** $G$ and $L_i$ indicate full caption and split captions (for our method), and indices denote diffusion steps (901 is closer to noise). White indicates high attention scores.

