# OpenReview forum: "Understanding Compositional Generalization via Hierarchical Concept Models"
_ICLR.cc/2026/Conference — ICLR 2026 Conference Withdrawn Submission_

### Official Review · Reviewer_CEJo · 2025-10-15

**Soundness:** 2
**Presentation:** 2
**Contribution:** 3
**Rating:** 4
**Confidence:** 3

**Summary:**

The paper's primary contribution is a novel theoretical framework for compositional generalization, which posits that this capability arises from a hierarchical decomposition of concepts, inspired by human analogical reasoning and formalized through causal modularity and minimal change principles. This framework accommodates complex, non-parametric interactions, advancing beyond the restrictive assumptions of prior work.

The authors provide theoretical guarantees for the identifiability of this latent structure from observational data and translate these insights into a practical implementation, HierDiff, for diffusion models. HierDiff operationalizes the theory by mapping conceptual hierarchies to diffusion timesteps and enforcing modularity via a novel sparsity regularization on cross-attention maps.

The effectiveness of this theoretically-motivated approach is demonstrated through performance improvements on the challenging DPG-Bench, validating the framework's utility and showcasing its scalability to larger models.

**Strengths:**

Originality:
The inspiration from human thought and implementation in real-world AI models is very original. Although I am not very familiar with the prior work, it seems like this work overcomes significant limitations and moves causal representation learning from theory to reality.

Quality & Clarity:
The writing quality and clarity are very good throughout the paper, as well as the analogies which help to give an intuitive understanding of the problem.

Significance:
The work moves causal representation learning from a theoretical concept/principle into a useful tool in the real world with real models, making it very significant in helping to find additional inductive biases to move us beyond "scaling is all you need"

**Weaknesses:**

1. Figure 1 has too many things going on and is very confusing. I think it would benefit from having less low-level concepts and arrows, as well as less colors, or some kind of legend - I am not sure what the dotted yellow, blue, and dotted orange circles are supposed to represent, or why some of them are dotted
2. The paper doesn't make it clear whether causal modularity and minimal-change are foundational concepts in causality, from prior work, or are introduced by the authors. I think some clarification on this would be helpful to the reader
3. In figure 3, it is unclear to me what d_1 and d_2 are (and what their dimensionality is). The figure indicates they are text, but are the y individual words? something more abstract? And the text mentions d_1,1 but it is unclear how this relates to d_1. Some clarification on this would be helpful to the reader
4. From what I understand, one key step in the process is decomposing the description into low-level text descriptions using a LLM. It seems like this is additional information to the model which the baselines are not given (although I am not familiar with the baselines). I think that comparing a baseline which uses a LLM to generate additional information related to the caption (but perhaps injects it in a less intelligent way) is necessary to make sure that the improvements in performance are not simply from the additional information provided by the LLM
5. Some discussion/comparison of the additional complexity/computation needed to train HierDiff compared to baselines should be included to see what are the tradeoffs of this increase in performance

**Questions:**

My questions and suggestions are detailed in the weaknesses section. I just want to make clear that I am very open to improving my score once all my concerns are addressed.

---

### Official Review · Reviewer_52zj · 2025-10-25

**Soundness:** 3
**Presentation:** 3
**Contribution:** 3
**Rating:** 4
**Confidence:** 4

**Summary:**

This paper tackles the fundamental challenge of ​compositional generalization for text-to-image generation. The author proposes HierDiff to enhance diffusion models with hierarchical concept conditioning (concepts provided by QWEN 2.5) instead of prompt conditioning and a sparse regularization to encourage sparse interactions among concepts. The theoretical outcomes are two principles called "causal modularity" and "minimal-change principle". The experimental results demonstrate that HierDiff outperforms other methods on handling complex prompts involving multiple concepts and relationships.

**Strengths:**

- The theoretical statements are clear and easy to follow. Unlike prior works that relied on restrictive assumptions, e.g., additive pr polynomial concept interactions, this work accommodates complex, nonparametric interactions among concepts within a hierarchical graphical model.
- The main idea to benefit diffusion models is by conditioning generation on concept embeddings instead of on the global text prompt. This is intuitive, and effective translation of theory into design.
- The experiments show that this method is successful scaling to a large diffusion transformer while maintaining competitive performance, which indicates that the theoretical insights are not architecture-specific but provide a generalizable blueprint for improving compositional generalization in large-scale models.

**Weaknesses:**

- While the paper presents a compelling theoretical framework for compositional generalization, several limitations and unaddressed questions persist, particularly regarding the specific impact of the proposed hierarchical concept model on text-to-image generalization performance.
- While the paper includes a "Related Work" section in the appendix, integrating it into the main body would immediately contextualize its novel theoretical contributions.
- The hierarchical data-generating process indicates a tree relationship on concepts. The proposed hierarchical graphical model could be simplified.
- Some notations are not consistent. In section 4, why not use $d$ for the text description and $z$ for embeddings as in section 3?
- The use of the language model to extract concepts from the high-level textual description may suffer from technical issues, like how to guarantee the correctness of such decomposition since a language model may suffer from hallucination.
- The current experimental design does not adequately validate the proposed hierarchical concept decomposition as a superior approach compared to alternative forms, despite this being a foundational motivation explicitly stated in the theoretical framework (lines 130-131).
- While the numerical differences between Table 2 and Table 3 appear marginal, the inclusion of supplementary statistics, such as standard deviation, is necessary to robustly substantiate the claimed performance improvements.

**Questions:**

- Based on my understanding, a tree model is enough for describing the hierarchical structure? What is the necessity to formulate it as a graphical model rather than a tree model?
- The notation in lines 175-176: $d_1$ and $d_{1,1}$ is weird for me. The relationship between $d_1$ and $d_{1,1}$ seems to be the absence and the appearance of 'peacocks'. Why are they not $d_1=0$ and $d_1=1$?
- What does the author mean by saying "hierarchical structures benefit compositional generalization" in lines 203-204? What is the relationship between this statement and the following key insight?
- What is the case without TD in the ablation studies? Is it just Stable Diffusion? "Using only the global text" means there is also no need to perform sparsity regularization on concepts for me.
- In line 450, is there a typo by saying "Relation from 84.83 to 83.03"? I do not find any "83.03" in Table 2. Please comment on any possible reason why HierDiff is worse than HierDiff-w/o TD on the cases of Global and Other.

---

### Official Review · Reviewer_ErGc · 2025-10-31

**Soundness:** 2
**Presentation:** 3
**Contribution:** 1
**Rating:** 2
**Confidence:** 3

**Summary:**

This work aims to understand how compositional generalization can be achieved using less restrictive assumptions than prior works. To this end, a hierarchical latent variable model for observed data is introduced along with the assumptions of causal modularity and minimal change. The authors argue theoretically that these assumptions provably enable compositional generalization. They then propose an empirical method based on their theory for diffusion models using an attention map regularizer and demonstrate improved compositionality relative to baseline models.

**Strengths:**

* The paper addresses an important problem, namely understanding how compositional generalization can be achieved in machines.

* The paper is relatively easy to read and understand.

* The paper contains many figures which are well made.

* The experimental results on image data in Section 5 using the proposed method appear promising.

**Weaknesses:**

### **Main weakness**
My main issue with this paper pertains to its purported theoretical contribution which makes up the majority of its contribution. The main claim made by the authors is that they can provably achieve compositional generalization using less restrictive assumptions than [1, 2, 3]. **I do not believe this theoretical claim to be true.**

The works of [1, 2, 3] define compositional generalization as the ability to recover the ground-truth generator function on the entire latent space by only observing a certain subset of those latents. If this is achieved, then images containing novel concept combinations can be generated. In order for this form of extrapolation to even be mathematically possible, the ground-truth generator must belong to a constrained function class [1,2, 3], hence the assumptions on constrained interactions from [1, 2, 3]. In the present work, as far as I understand, no constraints are placed on the generator's function class.


This on its own indicates to me that there is a misalignment between what the authors define as compositional generalization compared to prior works. My current understanding is that the authors are viewing compositional generalization as the ability to estimate the latent distribution out-of-domain, opposed to estimating the ground-truth encoder or decoder out-of-domain. Is this correct? If so, then this is a very different statement than those provided by [1,2,3] and does not constitute a meaningful compositional generalization result in my view.

### **Minor weaknesses**


* There are a few parts of the manuscript that I found needing further motivation. For example the sentence on line 134: “Humans understand and envision concept compositions through comparison and analogy..…?”,
should either be motivated further or include some sort of citation. Namely, given that this is the motivation for the authors main assumptions, I believe the claim should be substantiated further

* I also believe that it would be useful for the authors to relate their attention sparsity regularizer to the one proposed by Brady et. al 2024 [3], to better clarify the novelty/contribution here.


**References**


[1] Lachapelle et. al, 2023 Additive decoders for latent variables identification and cartesian-product extrapolation


[2] Wiedemer et. al, 2023 Provable Compositional Generalization for Object-Centric Learning


[3] Brady et. al, 2024, Interaction Asymmetry: A General Principle for Learning Composable Abstractions

**Questions:**

* How are the authors defining compositional generalization? How does this relate to the definitions in the prior works [1,2,3] that they aim to build upon?

* Are any assumptions on the ground-truth generator made apart from invertibility?

* How do the authors define interactions and how is this more flexible than the interactions used in Brady et. al 2024 [3]?

* What is the relationship between the regularizer in this work and the one proposed by Brady et. al 2024 [3]?

---

### Official Review · Reviewer_XyQw · 2025-11-01

**Soundness:** 2
**Presentation:** 3
**Contribution:** 2
**Rating:** 4
**Confidence:** 5

**Summary:**

The work studies compositional generalization in generative modeling. Compositional generalization is hard, and the paper views data in a hierarchical way, flowing from the text to lower-level concepts, which themselves combine into an image. The work brings results from identifiability theory and argues that one could achieve transfer to the full space of concept combinations: in particular, the work claims that identifiability is possible under a few assumptions, and as a consequence so is the transfer. The work then proposes a method based on the assumptions in the theory by using timestep weighting over the attention maps and low overlap between low-level concepts.

**Strengths:**

1. Motivation is strong, problem is relevant.
2. Approaching the problem from a hierarchical data‑generating‑process perspective is useful.

**Weaknesses:**

Major:
1. I don’t find the identifiability results particularly useful: the challenge, I believe, lies in matching the latent variables’ densities of the hierarchical data‑generating process. The theory says that if the model already matches the densities on the training supports __and__ it actually mimics the true data‑generating process, then the model should transfer to unseen combinations. But clearly there's no reason for the models to do so. This connects to the second weakness.
2. It’s not clear how the theory actually connects to the method. I understand the authors tried to justify this (l409–419), but, e.g., statements like “Invertibility (Cond. 3.3‑i): The model’s reconstruction objective inherently promotes invertibility” either require a reference or a clarification. More importantly, it’s unclear how this ties to the proposed losses. The “sparsity” reads more like a disentanglement prior.
- Only a single baseline was fine‑tuned: SD1.5. It’s unclear if the benefit transfers to other architectures, especially flow‑based approaches like SD3. I understand fine‑tuning is expensive, but so far I can’t tell if the results are specific to SD1.5 or not.
- Comparison with previous works is missing
	- The introduction opens with the question "What data structures enable compositional generalization, and how can we characterize them?", but this question isn’t answered concretely later. The “structure” part has been studied quite a bit already:  (1) In generative modeling, [1] showed that compositional generalization can emerge automatically under standard training on concept-disentangled data.  (2) In discriminative models,  [2] analyzed linear subspace structure (“linear spaces of meaning”) in VLMs, and [3] showed that such structures naturally emerge with scale and are directly linked to compositional generalization. Connecting to these directions would strengthen the story.

[1] Okawa et al. Compositional Abilities Emerge Multiplicatively: Exploring Diffusion Models on a Synthetic Task. arXiv:2310.09336.
[2] Trager et al. Linear Spaces of Meanings: Compositional Structures in Vision-Language Models. arXiv:2302.14383.
[3] Uselis et al. Does Data Scaling Lead to Visual Compositional Generalization? arXiv:2507.07102.

Minor:
- The title is too broad; the work studies comp‑gen in a generative setting. I also take issue with the “understanding” part - it’s not clear from the contributions what understanding was gained. To me, the method appears to be the main contribution.


I'm willing to increase my score if these weaknesses are addressed.

**Questions:**

1. l349: what is “information gap” in “information gap between zS(t+1) and zS(t)”?
2. Can simpler baselines be run? e.g., fine‑tuning on captions that include parts of caption information, or simply including parts of the hierarchy within the prompts?
3. Do the results transfer to different benchmarks? e.g., GenEval.

---

### Note · Authors · 2025-11-14

**Comment:**

We would like to sincerely thank the Area Chair and all the reviewers for their time and effort in reviewing our manuscript.

After careful consideration of the feedback, we have decided to withdraw our submission. We plan to improve our work for submission to a future venue, in light of the reviewers' valuable feedback.

We appreciate the opportunity to have submitted our work to ICLR.

**Withdrawal Confirmation:**

I have read and agree with the venue's withdrawal policy on behalf of myself and my co-authors.